# Reliability and Utility of Various Methods for Evaluation of Bone Union after Anterior Cervical Discectomy and Fusion

**DOI:** 10.3390/jcm11206066

**Published:** 2022-10-14

**Authors:** Bartosz Godlewski, Adam Bebenek, Maciej Dominiak, Marcin Bochniak, Piotr Cieslik, Tomasz Pawelczyk

**Affiliations:** 1Department of Orthopaedics and Traumatology, with spinal Surgery Ward, Scanmed—St. Raphael Hospital, ul. Adama Bochenka 12, 04-141 Cracow, Poland; 2Department of Orthopaedics and Traumatology, Military Institute of Medicine, 04-141 Warsaw, Poland; 3Department of Affective and Psychotic Disorders, Medical University of Lodz, 92-213 Lodz, Poland

**Keywords:** fusion assessment, cervical spine, ACDF, radiological measurements

## Abstract

Most surgical procedures performed on account of degenerative disease of the cervical spine involve a discectomy and interbody fixation. Bone fusion at the implant placement site is evaluated post-operatively. It is agreed that computed tomography is the best modality for assessing bone union. We evaluated the results obtained with various methods based solely on conventional radiographs in the same group of patients and compared them with results obtained using a method that is a combination of CT and conventional radiography, which we considered the most precise and a reference method. We operated on a total of 170 disc spaces in a group of 104 patients. Fusion was evaluated at 12 months after surgery with five different and popular classifications based on conventional radiographs and then compared with the reference method. Statistical analyses of test accuracy produced the following classification of fusion assessment methods with regard to the degree of consistency with the reference method, in descending order: (1) bone bridging is visible on the anterior and/or posterior edge of the operated disc space on a lateral radiograph; (2) change in the value of Cobb’s angle for a motion segment on flexion vs. extension radiographs (threshold for fusion vs. pseudoarthrosis is 2°); (3) change in the interspinous distance between process tips on flexion vs. extension radiographs (threshold of 2 mm); (4) change in the value of Cobb’s angle of a motion segment (threshold of 4°); (5) change in the interspinous distance between process bases on flexion vs. extension radiographs (threshold of 2 mm). When bone union is evaluated on the basis on radiographs, without CT evidence, we suggest using the “bone bridging” criterion as the most reliable commonly used approach to assessing bone union.

## 1. Introduction

Most surgical procedures performed on account of degenerative disease of the cervical spine involve a discectomy and interbody fixation. Post-operative assessment includes an evaluation of bone fusion at the implant placement site. A number of methods for evaluating bone fusion are in use. Some older approaches that are still in common use are based on conventional radiographs. Better and more reliable methods are based on CT images, CT being a more precise technique that allows for more objective results [1,2,3,4,5]. A previous paper of ours presented an evaluation of fusion at 12 months after surgery based on our own approach, which combines evidence from functional radiographs and CT scans [6]. In this paper, we assess and compare fusion results obtained with various methods based solely on conventional radiographs and then compare these results with the outcomes of our method, which we consider the most precise and a reference method. We also analyse changes in the radiographic indices assessed by comparing pre-operative values with those obtained at 12 months post-surgery. Most conventional methods based on post-operative radiographs do not take into account the pre-operative appearance/status. Criteria for classifying fusion are based exclusively on precise/specific values given in degrees or millimetres, and these underlie the classification of an outcome as fusion as opposed to pseudoarthrosis. Accordingly, our study also set out to verify whether the assessing pre-operative baseline status and subsequent changes is also advisable and whether they provide helpful parameters for classifying fusion.

## 2. Material and Methods

We operated on a total of 170 disc spaces in a group of consecutive 104 patients (age: 51.2 ± 10.3; female 73.1%) qualified for one- or two-level surgery. All patients were operated on by the same surgeon and according to the same technique. The procedures involved a discectomy, removal of osteophytes, transection of the posterior longitudinal ligament and decompression of neural structures, followed by the insertion of an interbody implant (cage), whose interior was always filled with nanoparticle hydroxyapatite. Fusion was evaluated at 12 months after surgery with 5 different and popular classifications based on conventional radiographs [7,8,9,10,11,12,13,14]:change in the distance (in millimetres) between the bases of spinous processes in flexion vs. extension. Values ≥2 mm are considered a sign of fusion while values <2 mm represent a pseudoarthrosis.change in the distance (in millimetres) between the tips of spinous processes in flexion vs. extension. The threshold values are the same as with the 1st method above.change in the value of Cobb’s angle of the operated motion segment in flexion vs. extension. Changes <2° are a sign of fusion while changes ≥2° represent a pseudoarthrosis.change in the value of Cobb’s angle of the operated motion segment in flexion vs. extension, as in the 3rd method above, but with a different threshold: <4° is fusion and ≥4° is a pseudoarthrosis.bone bridging visible on the anterior and/or posterior edge of the operated disc space in a lateral radiograph. Fusion is diagnosed when bone bridging is visible, and if it is not visible, a pseudoarthrosis is diagnosed.

These approaches to assessing fusion are presented in Figure 1, Figure 2 and Figure 3.

Measurements were made on radiographs obtained in one X-ray centre following the same procedure and utilising the same equipment and software. Analysis of both conventional radiographs and CT images was performed by 3 individuals (2 neurosurgeons, 1 orthopaedist) who were also among the authors of the paper (Bartosz Godlewski, Adam Bebenek and Maciej Dominiak). The radiographic studies were analysed jointly and a final assessment was made (numerical value was entered) taking into account the opinion of each participant.

The results were subjected to a statistical analysis and compared. The radiograph-based results were also compared to an evaluation concerning the same group of patients but performed according to our reference method that combines radiographic and CT evidence, treating this method as more precise and a point of reference. As this method relied on combined evaluation of CT scans and radiographs, the final sample comprised 144 disc spaces. Complete fusion was demonstrated in 101 cases (71.1%) and partial fusion, in 43 cases (29.9%). There were no cases that could be considered absence of fusion. The classification criteria for the reference method are presented in Table 1. Examples of cases classified as complete fusion and partial fusion (pseudoarthrosis) on CT scans are shown in Figure 4 [6]. For the analysis of the radiograph-based methods involving changes in angles and distances in flexion vs. extension, we additionally compared changes in these indices between the pre-operative baseline and the 12-month post-op status. The research was approved by the Bioethics Committee of the Andrzej Frycz Modrzewski University in Cracow (Resolution 4/2019) and was conducted in compliance with the Declaration of Helsinki.

### Statistical Methods

An analysis of the indices used in the diagnostic tests was conducted with the Jamovi statistical software package (version 2.2.5.0) to determine the degree of concordance between these five approaches and the reference method. As the variables were qualitative, 2 × 2 contingency tables were used in the analysis for sensitivity, specificity, AUC and accuracy. Test accuracy, i.e., the measure of a diagnostic test that assesses its ability to produce true (i.e., consistent with the reference method) results, was adopted as the basis for assessing the concordance of individual approaches with the reference approach. The statistical calculations were performed in keeping with the Standards for Reporting of Diagnostic Accuracy (STARD) [15], with 95% confidence intervals for statistical accuracy calculated in the tables.

## 3. Results

The best fit with the reference method was obtained for the approach that looked for the presence of bridging bone. The percentages of complete fusion vs. pseudoarthrosis obtained with the individual approaches are presented in Figure 5.

The following ranking of the radiographic approaches with regard to their fit with the reference method, in descending order, was formed following the statistical analyses on the basis of test accuracy:Method E—(Bridging bone).Method D—(Cobb’s angle with 4° threshold).Method B—(Interspinous distance at tip).Method C—(Cobb’s angle with 2° threshold).Method A—(Interspinous distance at base).

See Table 2 for the detailed results.

A comparison of the methods based solely on post-operative radiographs revealed differences between the results even though the data were obtained from the same group of patients, which shows disadvantages of these approaches. Accordingly, the presence of fusion vs. a pseudoarthrosis, in our opinion, should not be assessed with these methods. Radiograph-only assessment is more subjective, less reproducible and has greater interobserver variability [6,7,16,17], as evidenced by the data in Table 2.

The differences between the results may also have been due to the assessors strictly following the threshold values for changes in distances and angles. We had supposed that of importance for assessing fusion could also be the extent of changes in distances and angles between pre-operative and 12-month post-operative measurements and that assessment should not be based solely on post-operative appearance while ignoring pre-operative status. Accordingly, we performed measurements of flexion/extension mobility before and 12 months after the surgery, and subsequently calculated differences in changes of the distances and angles between these time points. We compared these results with regard to their concordance with the reference method. Next, we used receiver operating characteristics (ROC) and Youden’s J index [18] to calculate cut-off points (differences in angles or distances between flexion and extension) corresponding to maximum values of the discriminating indices (sensitivity and specificity). To this end, with significant values of the area under the curve for ROC, we calculated the change criteria maximizing sensitivity and specificity values using Youden’s J index. These results are shown in Table 3. Significant AUC values for ROC were obtained for the difference between differences in Cobb’s angles for flexion and extension (R_R_COBB) before surgery and at 12 months post-surgery (*p* = 0.021). Youden’s J index values indicate that discrimination for the new method employed in our study was maximized for >1.89. With this criterion, the sensitivity and specificity values reached 63.98% and 61.76%, respectively. Calculated pre- vs. post-surgery differences between differences in distances for flexion and extension did not generate significant AUC values for ROC (see Table 3). The outcomes of measurements based on differences in the difference between angles and distances in flexion vs. extension pre-operatively at 12 months post-operatively had significant discriminating power compared to the reference CT-based method only with regard to angles. Considering all statistical calculations, we believe that methods based on analysing changes in angles and distances cannot be regarded as more reliable than those based solely on post-operative radiographic images.

## 4. Discussion

It is generally agreed that computed tomography with image reconstruction is the best method for assessing bone fusion. This approach is more precise and reliable than methods based solely on conventional radiographs [1,2,3,4,5]. Ploumis et al. found a greater degree of interobserver concordance for CT-based classifications (89% for CT vs. 81% for radiographs). They state that the average fusion rate assessed by CT was lower (74%) compared to radiograph-based approaches (81%). In their paper, CT assessment led to higher pseudarthrosis rates than plain radiographs: 13 to 31% according to CT; 2 to 16% according to plain radiographs. The difference averaged 11%. Consistency between reviewers was higher with CT (average agreement: 89%; range 82–96%) than with plain radiographs (average agreement: 81%; range: 76% to 87%) [3]. Skolasky et al. found assessment consistency of just 54% between a panel of independent experts and the treating surgeon for radiograph-based evaluation of the same group of patients. The surgeon was more likely to report fusion than the independent panel of experts [19]. In order to improve assessment reliability, it seems reasonable to use approaches that additionally involve CT evidence. Buchowski et al. performed cervical spine revision surgeries and compared intraoperative findings with pre-operative X-ray, CT and MRI results. They found that it was CT that agreed most closely with intraoperative findings compared to plain radiographs or MRI scans [9]. Cannada et al. studied the accuracy and reliability of methods based on changes in Cobb’s angle and interspinous distance, finding that classifications of bone fusion based on changes in interspinous distances in flexion vs. extension were more accurate than Cobb’s angle measurements [20]. There are reports showing that assessment based on bone trabeculation/ bone bridging on static radiographs should be regarded as less reliable than assessment based on functional radiographs (flexion/extension) [8,9]. Our current results contradict those conclusions. Although MRI scans can also be used to evaluate fusion, the magnetic susceptibility artefact makes this modality less reliable than CT scans [9,21,22,23,24,25]. In a previous article of ours, the presence of fusion was assessed according to our original method based on the combined interpretation of radiographs and CT scans [6]. In the present paper, this method is treated as a reference, and its outcomes as the most reliable. We compared several of the most popular radiograph-based methods against our approach. The greatest concordance with the reference method was obtained for the approach based on the presence of bone bridging on the anterior and/or posterior edge of the operated disc space on a lateral radiograph. The presence of bridging bone corresponds to fusion, and absence is classified as a pseudoarthrosis. Accordingly, in cases when the presence of fusion is assessed on the basis of radiographs and CT scans are not available, we recommend using bone bridging visible on radiographs as evidence for fusion as the most reliable method among several popular approaches. The outcomes of measurements based on differences in the difference between angles and distances in flexion vs. extension pre-operatively at 12 months post-operatively had significant discriminating power compared to the reference CT-based method only with regard to angles. The discrimination criterion based on Youden’s J index indicated maximum sensitivity and specificity for a criterion of >1.89. However, the values of sensitivity (63.89%) and specificity (61.76%) for this approach are not superior to those (69.5% and 69.2%, respectively) calculated for the radiograph-only method based on the presence of bridging bone, which does not allow for recognizing the method involving pre- vs. post-operative functional measurements as diagnostically more consistent with the reference method than the radiograph-based approach assessing the presence of bridging bone. Considering all statistical calculations, we believe that methods based on analysing changes in angles and distances cannot be regarded as more reliable than those based solely on post-operative radiographic images.

## 5. Conclusions

CT analysis represents the best approach to assessing the presence of bone fusion and should continue to be regarded as a gold standard. Among radiograph-based methods, the assessment of bone bridging on the anterior and/or posterior edge of the operated disc space on a lateral radiograph emerges as the most reliable method showing the greatest degree of consistency with CT-based evaluation. The presence of bone bridging is interpreted as fusion and its absence is interpreted as a pseudoarthrosis. We suggest using this method as the most reliable among popular radiograph-based methods for assessing fusion on the basis of radiographs when a CT scan is not available.

## Figures and Tables

**Figure 1 jcm-11-06066-f001:**
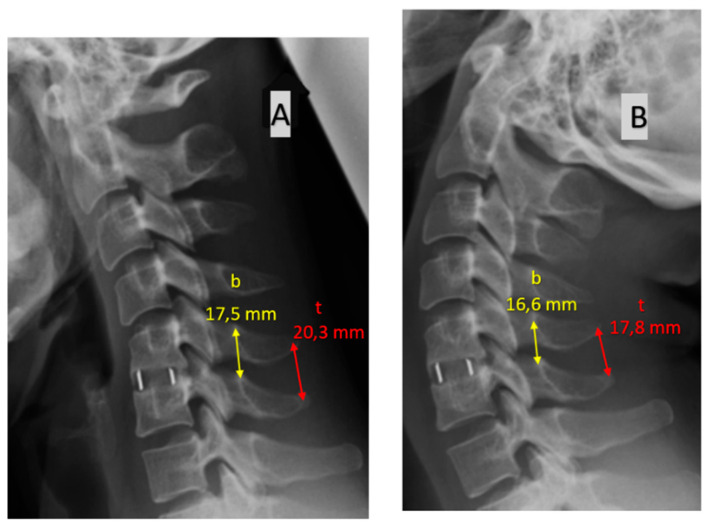
Assessment of bone fusion based on the change in the distance (in mm) in flexion (**A**) vs. in extension (**B**) between the bases of spinous processes (b) and between the tips of spinous processes (t). Greater interspinous distances are noted in flexion.

**Figure 2 jcm-11-06066-f002:**
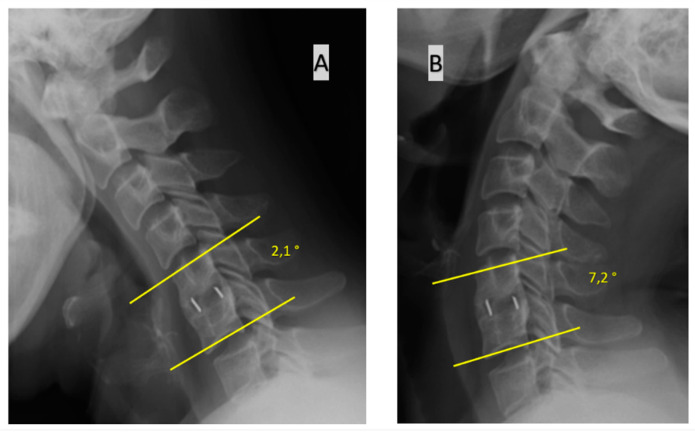
Assessment of bone fusion based on the change of Cobb’s angle of the operated motion segment in flexion (**A**) vs. in extension (**B**). Lower Cobb’s angles are noted in flexion.

**Figure 3 jcm-11-06066-f003:**
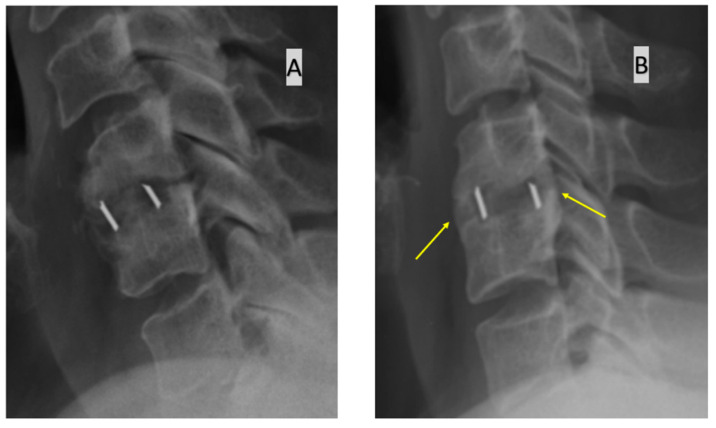
Assessment of bone fusion based on the identification of bone bridging on the anterior and/or posterior edge of the operated disc space. (**A**) radiograph at 12 months after surgery without bone bridging (**B**) radiograph at 12 months after surgery with bone bridging present (arrows).

**Figure 4 jcm-11-06066-f004:**
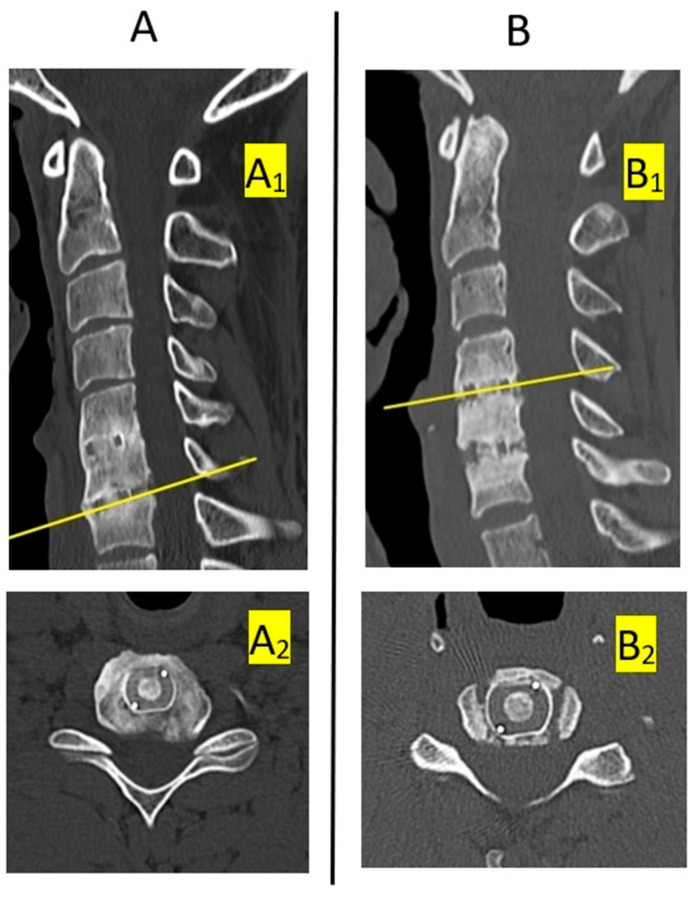
Sample presentations of complete and partial fusion on CT scans at 12 months post-surgery: (**A**) complete fusion (A_1_—sagittal view, A_2_—transverse view at the level of the implant in C6/C7 disc space). (**B**) partial fusion (B_1_—sagittal view, B_2_—transverse view at the level of the implant in C4/C5 disc space). Reprinted by permission from Springer Nature: Acta Neurochirurgica 2022, 164 (6), 1501-1507. PEEK versus titanium-coated PEEK cervical cages: fusion rate. Godlewski B, Bebenek A, Dominiak M, Karpinski G, Cieslik P, Pawelczyk T.

**Figure 5 jcm-11-06066-f005:**
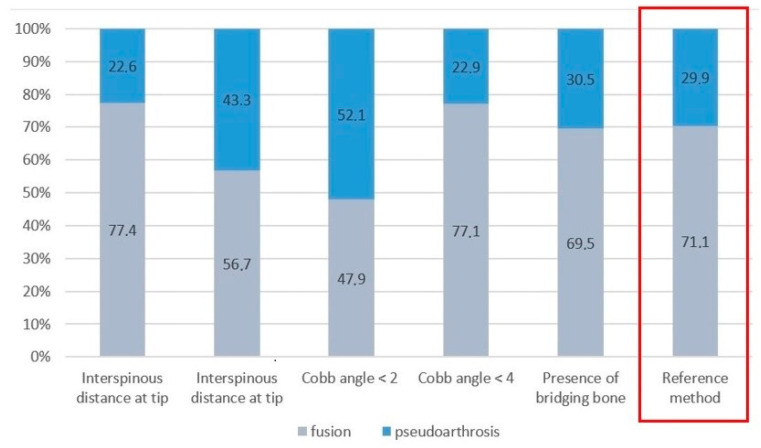
Fusion rates in percentages according to various assessment methods.

**Table 1 jcm-11-06066-t001:** Criteria for evaluation of fusion based on CT scans and functional conventional radiographs of cervical spine at 12 months post-surgery. Reprinted by permission from Springer Nature: Acta Neurochirurgica 2022, 164 (6), 1501–1507. PEEK versus titanium-coated PEEK cervical cages: fusion rate. Godlewski B, Bebenek A, Dominiak M, Karpinski G, Cieslik P, Pawelczyk T.

Modality	Criterion	Complete Fusion	Partial Fusion	Absence of Fusion
**Functional radiographs**	Mobility of implants against vertebral bodies on functional radiographs	No mobility	No mobility	Visible mobility
**Computed tomography images**	Continuity of bone tissue immediately anterior, posterior, medial and lateral to implant on CT scan	Visible bone tissue continuity	No continuity of bone tissue	No continuity of bone tissue

**Table 2 jcm-11-06066-t002:** Approaches to assessing fusion after 12 months in relation to fusion status assessed in the current study.

Diagnostic Approach	Sensitivity ^a^ [%]	Specificity ^b^ [%]	AUC	Test Accuracy ^c^ [%]	95% CI for Accuracy	Accuracy Rank
Cobb Angle 2°	47.9	53.7	0.51	49.6	40,987–58,299	4
Cobb Angle 4°	77.1	24.4	0.51	61.3	52,621–69,507	2
Interspinous distance at base	22.6	53.8	0.38	31.8	23,987–40,486	5
Interspinous distance at tip	56.7	66.7	0.62	59.7	50,696–68,229	3
Presence of bridging bone	69.5	69.2	0.69	69.4	60,859–77,067	1

Note: ^a^ Sensitivity (True positives among diseased). ^b^ Specificity (True negatives among healthy). ^c^ Accuracy (true test result ratio)—the ability of the test to reveal true results. AUC—area under the ROC curve. CI—confidence interval.

**Table 3 jcm-11-06066-t003:** Receiver operating characteristics (ROC) curve analyses for the reference method and parameters based on differences between flexion and extension measured before and 12 months after the surgical procedure.

Variable	Area under the ROC Curve (AUC)	95% CI for AUC	*p* Value	Youden J Index	Associated Criterion	Sensitivity	Specificity
Difference in angles (R_R_Cobb)	0.629	0.529–0.722	0.021	0.257	>1.89	63.89	61.76
Difference in interspinous distance at base (R_R_Base)	0.532	0.431–0.631	0.609	-	-	-	-
Difference in interspinous distance at tip (R_R_Tip)	0.529	0.426–0.630	0.646	-	-	-	-

Note: CI—confidence interval, significant results are underlined.

## Data Availability

Data supporting results of this study can be assessed on request to the corresponding author.

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
