# Peer review of "Reliability and Utility of Various Methods for Evaluation of Bone Union after Anterior Cervical Discectomy and Fusion"

_jcm, 2022, doi:10.3390/jcm11206066_

Round 1
Reviewer 1 Report
This is a good job, meticulously done. However, I think the authors should be more cautious in their conclusions. In Material and Methods section, they do not refer to the design of the study, so we do not know what the sample size would be. On the other hand, the statistical analyses show very borderline values, both AUC and Sensitivity and Specificity. I insist that they should be more careful when issuing their conclusions.
Author Response
Reply to suggestions and questions from Reviewer #1:
Thank you for your review and comments on the manuscript.
Question/suggestion 1: “This is a good job, meticulously done. However, I think the authors should be more cautious in their conclusions. In Material and methods section, they do not refer to the design of the study, so we do not know what the sample size would be. On the other hand, the statistical analyses show very borderline values, both AUC and Sensitivity and Specificity. I insist that they should be more careful when issuing their conclusions”
Answer 1: We have amended the paper as suggested. We have added information under „Material and methods”. In the Results, Discussion and Conclusions sections, we have now mitigated our previous opinions considering the statistical data and the reviewer’s recommendations.
Reviewer 2 Report
The authors compared 5 well established methods for evaluation of interbody fusion in cervical spine treated for degenerative disease. The authors compared these imaging methods on a total of 170 disc spaces in a group of 104 patients. They concluded that when bone union is evaluated on the basis on radiographs, without CT evidence, they recommend using the “bone bridging” criterion as the most reliable commonly used approach to assessing bone union.
Interesting paper with appropriated statistical analysis. Were al 170 discs in 104 consecutive patients? Or selected on another basis?
“Fusion was evaluated at 12 months after surgery” Why not at 6 months or 18 months
Why sliding of the vertebra postoperatively was not included in the roentgenographic analysis since it is usually considered as sing of pseudarthrosis.
Why not focal osteolysis around the peek or other intervertebral devise were not taken in consideration as sing for pseudarthrosis?
“Ploumis et al. found a greater de-181 gree of interobserver concordance for CT-based classifications (89% for CT vs 81% for 182 radiographs).” And “Measurements were made on radiographs obtained in one X-ray centre, following 89 the same procedure and utilising the same equipment and software” Who made the roentgenographic evaluation ? How many observers? Interobserver/intraobserver agreement?
“For the analy-99 sis of the radiograph-based methods involving changes in angles and distances in flexion 100 vs extension, we additionally compared changes in these indices between the preoperative baseline and the 12-month post-op status”. Interobserver/intraobserver agreement is required.
Discussion too short. Add more related literature.
Author Response
Reply to suggestions and questions from Reviewer #2:
Thank you for your review and comments on the manuscript.
Question/suggestion 1: “Interesting paper with appropriated statistical analysis. Were all 170 discs in 104 consecutive patients or selected on another basis?”
Answer 1: The procedures were performed in 104 consecutive patients who had been qualified for one- or two-level surgery. Patients requiring ≥ 3 disc level surgery were not included in the analysis.
Question/suggestion 2: “Fusion was evaluated at 12 months after surgery”, “Why not at 6 months or 18 months?”
Answer 2: The 12-month period appears optimal for evaluation of fusion. At the 12-month mark, there should be observable radiological evidence of complete fusion. Investigating fusion at 6 months appear to be too early. The evaluation could naturally have taken place at 18 months, but we considered 12 months to be an optimal period and so adopted our methodology.
Question/suggestion 3: “Why sliding of the vertebra postoperatively was not included in the roentgenographic analysis since it is usually considered as sing of pseudoarthrosis?”
Answer 3: Implant subsidence after ACDF is an undesirable effect that should be prevented. We agree that the optimal radiographic outcome following ACDF is complete fusion without implant subsidence. The finding of cage subsidence does not, however, preclude the possibility of complete fusion at the implant placement site later on. Even if endplate continuity is broken at an early stage and the implant subsides towards a neighbouring vertebral body, complete fusion can still be achieved around the implant in the longer term. That was the reason why we did not account for implant subsidence in the final analysis of fusion. Our previous study [Godlewski B, Bebenek A, Dominiak M, Karpinski G, Cieslik P, Pawelczyk T. PEEK versus titanium-coated PEEK cervical cages: fusion rate. Acta Neurochir (Wien). 2022, 164(6), 1501-1507. doi: 10.1007/s00701-022-05217-7 demonstrated that there was no association between the type of fusion and the presence of implant subsidence (B = 0.461; P = 0.2903). The 101 levels assessed as complete fusion included 18 (17.8%) cases of subsidence and 83 (82.2%) cases without subsidence. Thus, optimal treatment outcome (complete fusion without implant subsidence) was obtained in 83 of the 144 disc spaces analysed, which represents 57.6%.
Question/suggestion 4: “Why not focal osteolysis around the peek or other intervertebral device were not taken in consideration as sing for pseudarthrosis?”
Answer 4: We agree that this can be a sign of pseudoarthrosis, but for our purposes we had to choose one method for evaluating fusion that would be the best with regard the issue analysed. We decided to adopt the method described in the paper, based on both conventional radiographs and CT with the criteria for classification described in detail there.
Question/suggestion 5: “Ploumis et al. found a greater de-181 gree of interobserver concordance for CT-based classifications (89% for CT vs 81% for 182 radiographs).” “Who made the roentgenographic evaluation ? How many observers? Interobserver/intraobserver agreement?”
Answer 5: In the paper by Ploumis et al. four independent blinded observers evaluated the roentgenological data. Unfortunately, the study failed to describe how the interpreters were “blinded”. In that paper, CT assessment led to higher pseudarthrosis rates than plain radiographs: 13 to 31% according to CT; 2 to 16% according to plain radiographs. The difference averaged 11%. Consistency between reviewers was higher with CT (average agreement: 89%; range 82%-96%) than with plain radiographs (average agreement: 81%; range: 76% to 87%).
Question/suggestion 6:
“Measurements were made on radiographs obtained in one X-ray centre, following 89 the same procedure and utilising the same equipment and software” Who made the roentgenographic evaluation ? How many observers? Interobserver/intraobserver agreement?”
“For the analy-99 sis of the radiograph-based methods involving changes in angles and distances in flexion 100 vs extension, we additionally compared changes in these indices between the preoperative baseline and the 12-month post-op status”. Interobserver/intraobserver agreement is required”
Answer 6: Analysis of both conventional radiographs and CT images was performed by 3 individuals (2 neurosurgeons, 1 orthopaedist) who were also among the authors of the paper (Bartosz Godlewski, Adam Bębenek and Maciej Dominiak). The radiographic studies were analysed jointly and a final assessment was made (numerical value was entered) taking into account the opinion of each participant. We have added relevant information in the manuscript.
Question /suggestion7: „Discussion too short”, „Add more related literature”
Answer 7: We have added more information to the discussion, based on recent literaturę, and we have added more references.
Round 2
Reviewer 1 Report
I consider that the authors' modifications are sufficient to proceed with the acceptance and subsequent publication of the work.